# Analyzing the impact of spatial centrality and courtyard diversity on tourist attractions in the walled city of Lefkoşa

Mustafa Aziz Amen [1]*, Farhan Abdullah Ali[2]

1 Architecture Department, The American University of Kurdistan, Duhok, Iraq, 2 Architecture Department, University of Kurdistan Hewlêr, Erbil, Iraq

* mustafa.amen@auk.edu.krd

## Abstract

The Walled City of Lefkoşa faces the dual challenge of preserving its historic urban fabric while addressing the evolving demands of modern tourism. A key problem lies in understanding the interplay between spatial centrality within the urban street network and the diversity of courtyard activities influencing tourist attraction and engagement. This study hypothesizes that while spatial centrality plays a significant role in guiding tourist movement and enhancing visibility, the diversity and adaptability of courtyard spaces have a greater impact on enriching visitor experiences. To test this hypothesis, the research employs a mixed-methods approach, combining quantitative and qualitative analyses. PageRank and Straightness centrality analysis are used to model the street network and assess the prominence and connectivity of key streets. Simultaneously, field observations and visitor data are integrated into GIS-based mapping tools to visualize spatial patterns of movement and activity. The findings reveal that courtyards hosting diverse and adaptable activities significantly enhance tourist engagement, even in less central locations. In contrast, areas with high centrality but limited courtyard adaptability struggle to sustain prolonged visitor interactions. These results underscore the importance of integrating spatial centrality with thoughtfully designed, multifunctional courtyard spaces in urban planning strategies. This research offers actionable insights for urban planners, architects, and heritage managers, advocating for a balanced approach that preserves the historic character of urban areas while meeting the dynamic needs of contemporary tourism.

## Introduction

The Walled City of Lefkoşa, a UNESCO heritage site, represents a unique blend of historical architecture, cultural vibrancy, and evolving urban dynamics [1]. As a historic urban landscape, Lefkoşa faces the dual challenge of preserving its rich heritage while adapting to the growing demands of modern tourism. The spatial centrality and

**Data availability statement:** All relevant data are within the manuscript and its Supporting Information files All relevant data are available from the Zenodo repository under DOI: https://doi.org/10.5281/zenodo.15571365

**Funding:** The author(s) received no specific funding for this work.

**Competing interests:** The authors have declared that no competing interests exist.

the visibility of key urban nodes have often been highlighted as drivers of the tourist movement, but the role of urban courtyard diversity and adaptability in attracting and engaging tourists has received relatively little attention.

This study aims to address this gap by exploring how the interplay between spatial centrality, shaped by the urban street network, and the diversity of courtyard activities impacts tourist attraction in Lefkoşa. This study addresses two central research questions:

(1) How does spatial centrality within the street network affect tourist distribution and movement in Lefkoşa's old city?

(2) How does the diversity of courtyard activities influence tourist engagement across varying levels of spatial centrality?

To answer these questions, the study adopts a mixed-methods analytical framework integrating GIS-based network analysis (PageRank and Straightness centrality) with field observations and regression modeling, offering a spatial and behavioral understanding of urban tourist dynamics.

Spatial centrality has long been recognized for its ability to direct pedestrian movement and increase the prominence of urban spaces [2–4]. Yet, this research posits that the adaptability and functional variety of courtyard spaces have a more profound influence on visitor engagement.

Courtyards that support various activities, encourage social interactions, and provide flexible spaces become attractive destinations for tourists, regardless of their location in the city.

To test this hypothesis, the study employs a mixed-methods approach combining quantitative and qualitative analyses. PageRank and Straightness centrality analysis are used to model the street network and evaluate the prominence and connectivity of key streets within the city. This analysis is complemented by field observations and the collection of visitor data, which are integrated into GIS-based mapping tools to visualize patterns of movement and courtyard usage.

## Literature review

Centrality is a fundamental concept in urban design, crucial for understanding patterns of movement, accessibility, and visibility within cities. [5–11]. Spatial centrality, often quantified through network analysis tools such as Closeness, betweenness, PageRank, evaluates the relative importance of nodes within a street network based on pedestrian flow and connectivity [12]. The centrality affects the movement patterns. for example, Hillier and Hanson's (1984) Space Syntax theory highlights how spatial configurations influence movement and social interactions, asserting that highly central locations naturally attract greater visibility and foot traffic [13–15]. This framework has been extensively applied in urban studies to examine pedestrian behavior, commercial activity, and tourism flows [4,16–21].

Chiaradia et al. (2012) explored how network centrality supports the clustering of activities in urban spaces, fostering social engagement and economic vitality [22].

Scholars contribute to this body of knowledge by integrating visibility graph analysis (VGA) and GIS to evaluate spatial patterns in heritage cities, offering insights into how centrality influences pedestrian movement in historically significant urban contexts, also emphasizing the role of architectural features, such as courtyards, in enhancing connectivity and spatial coherence in urban heritage sites, bridging a critical gap in the literature [23]. In the same field, Wen et al. (2023) developed a method to evaluate community tourism centrality based on network analysis and examined the correlation between community tourism centrality, tourism, and commercial gentrification [24]. Despite these advancements, much of the literature focuses predominantly on the role of street networks in facilitating movement, often overlooking how architectural features and the internal spatial diversity of specific areas, such as courtyards, interact with centrality to influence urban dynamics.

This gap in research highlights the need to examine how centrality interacts with the spatial and functional qualities of courtyards. Courtyards, often located within dense urban fabrics, provide unique opportunities to explore how centrality supports micro-scale urban dynamics, including informal social interactions, thermal comfort, and cultural expression [25]. Integrating centrality measures with analyses of courtyard morphology and functionality could offer a more comprehensive understanding of how urban spaces shape human behavior and movement patterns in heritage and modern contexts alike.

### PageRank urban centrality

While PageRank originated in web analysis and found applications in fields such as epidemiology and bioinformatics, its relevance to urban planning lies in its ability to assess indirect influence in complex networks. In urban studies, PageRank has been used to identify critical nodes in transportation and street networks, offering insights into pedestrian flows and accessibility [12,26,27].

PageRank centrality has emerged as a transformative tool in urban planning, offering a nuanced approach to analyzing and optimizing the interconnected networks that define modern cities [28,29]. Unlike traditional centrality measures, which often focus solely on direct relationships or the number of connections a node possesses, PageRank centrality evaluates the influence of a node by considering both direct and indirect interactions across the entire network. This layered perspective makes it uniquely suited for understanding and managing the complexities of urban systems, where the interactions between infrastructure, services, and people are highly dynamic and interdependent. [12]

One of the most critical contributions of PageRank centrality in urban planning lies in its ability to identify high-priority nodes within urban networks. Transportation hubs, utility grids, emergency service centers, and communication networks often form the backbone of city functionality [30]. In transportation, the algorithm's ability to unravel the structure of networks found a natural application in transportation systems. Representing intersections as nodes and roads as edges, researchers used PageRank to pinpoint vital transit hubs and critical roadways [28,31–33]. Moreover, strengthening key transportation hubs or decentralizing power grids based on PageRank insights can minimize the cascading effects of disruptions during emergencies [31,34].

### Straightness centrality

Straightness centrality plays a crucial role in urban planning by assessing the efficiency and directness of paths in a city's transportation and spatial networks [3,35–38]. This measure evaluates how closely the actual path between two points approximates the straight-line distance, providing valuable insights into the ease of movement across an urban area [39,40]. High straightness centrality is associated with well-connected urban spaces where destinations are accessible with minimal detours, thereby enhancing mobility for residents, reducing travel time, and improving the overall efficiency of transportation networks [26,41].

We chose to focus on straightness centrality in urban planning because it is a robust and versatile metric that directly reflects the efficiency and accessibility of a city's transportation and spatial networks [35–37]. Straightness centrality quantifies how direct the paths between locations are compared to the shortest possible distance, making it a key indicator of

spatial organization and movement potential within urban environments. This measure is particularly relevant in addressing critical urban challenges, such as reducing travel times [37], improving connectivity [3]and enhancing accessibility to essential services and amenities [42]. Straightness centrality also aligns with broader urban planning objectives, including sustainability, equity, and resilience [42]. By emphasizing shorter, more direct routes, it supports the promotion of low-carbon transportation modes such as walking and cycling, which are essential for reducing urban emissions and fostering environmentally friendly cities. Furthermore, the metric highlights areas with poor connectivity, helping planners address spatial inequities by improving access to underserved neighborhoods. This makes straightness centrality a powerful tool not only for optimizing urban layouts but also for ensuring inclusive development that benefits all segments of the population.

PageRank was selected for its ability to identify not only directly connected nodes but also those that are indirectly influential through the broader network, capturing hidden patterns of pedestrian importance often missed by simple degree or closeness measures [12,26]. In dense urban heritage settings like Lefkoşa, this capacity is crucial because street visibility and prominence are shaped by both immediate and global network positions. Straightness centrality, on the other hand, reflects the geometric efficiency of paths and aligns well with pedestrian preferences for direct and intuitive routes, especially in walkable environments [40]. Unlike betweenness, which emphasizes through-traffic, or closeness, which favors centralized locations, Straightness offers insights into perceived spatial legibility, a key factor in how tourists navigate historical areas.

## Courtyards as social and functional spaces

Courtyards have historically been integral to urban design, particularly in Mediterranean and Middle Eastern cities, where they blend architectural aesthetics with social and functional roles. Beyond their architectural significance, courtyards serve as venues for community interaction, cultural exchange, and various activities, and play a significant role in attracting tourism.

Ibrahim et al (2021) emphasized the importance of courtyards in educational buildings, noting that well-designed courtyards foster social sustainability by enhancing occupant interaction and contributing to a vibrant community atmosphere. Their study found that courtyards with flexible designs—accommodating cultural events, markets, or rest areas—contribute to the overall social and physical integration of urban environments. These characteristics make courtyards attractive to both locals and visitors, increasing their role as urban tourism assets [43].

Also, scholars highlighted the role of traditional courtyards in cultural sustainability. This adaptability makes them appealing not only to residents but also to tourists seeking immersive cultural experiences [44]. Chen et al. (2024) explored heritage regeneration models for traditional courtyard houses in Jinan, China, within the context of urban renewal. Their research emphasizes the importance of collaborative efforts among various stakeholders to optimize the renewal mechanisms of urban heritage, thereby enhancing the cultural and aesthetic appeal of courtyards. Such regenerated courtyards can become focal points for cultural tourism, attracting visitors interested in heritage and history [45]. Additionally, a study on sustainable urban landscapes in hot–dry regions examined the role of courtyards in microclimate regulation and enhanced thermal comfort.

## Materials and methods

To investigate the impact of spatial centrality and courtyard diversity on tourist attraction in the Walled City of Lefkoşa, this study employs a mixed-methods approach. The methodology integrates **quantitative spatial analysis** with **qualitative assessments** of architectural and social dynamics. This dual approach ensures a comprehensive understanding of how the spatial layout of the city and the functional diversity of its courtyards influence tourist engagement. Fieldwork and site documentation for this study were approved by the Department of Architecture at Girne American University, Northern Cyprus, with formal academic oversight and permission granted by the course instructor, Prof. Jose Madrigal. As the

research involved spatial analysis and visual documentation of publicly accessible urban areas within the Walled City of Lefkoşa, no government-issued permits were required. All activities were conducted in accordance with institutional guidelines and local norms regarding public space research.

The research focuses on the historical urban landscape of Lefkoşa, characterized by its rich architectural heritage and unique urban morphology. The study area was mapped and divided into zones based on spatial accessibility, architectural historical landmarks. Data collection was designed to capture the interplay between spatial centrality and tourist behavior. The study relied on two primary datasets: spatial network data, architectural data on courtyards, and tourist behavior data. The data for this research are available at https://doi.org/10.5281/zenodo.15571365

### Spatial network data

The spatial layout of the Walled City was digitized using GIS software to model the street network as a connected graph. Streets were represented as edges and intersections as nodes, capturing attributes such as connectivity, width, and surface type. Maps and historical plans from local municipalities and heritage organizations were verified through satellite imagery and field surveys. Informal pathways and hidden pedestrian routes were also incorporated to ensure accuracy.

### Tourist behavior data

Observational studies were conducted at key intersections and courtyards to record movement patterns and activities. Visitor counts were systematically carried out at different times of the day (morning, midday, and afternoon) on both a weekday (Monday) and a weekend (Sunday). Automated counters complemented manual counting to ensure accuracy. Behavioral indicators such as dwell time, engagement levels, and activity diversity were documented. Semi-structured interviews with tourists, local vendors, and guides provided qualitative insights into how visitors perceive and engage with courtyards.

A key component of the research involves a detailed analysis of the study area where data was collected. The investigation took place in the Turkish-administered section of Nicosia's old city, an area renowned for its historical and cultural richness. This part of the city is celebrated for its architectural legacy, prominent landmarks, and administrative significance, making it a focal point for tourism within Cyprus. Notable landmarks in the area include The Büyük Han, a restored Ottoman-era caravanserai; Bedesten, a historical marketplace repurposed as a cultural venue; and the Venetian Column, a striking relic of colonial history. These sites not only hold historical value but also serve as vibrant social and cultural hubs, offering visitors access to entertainment, dining, and leisure activities.

To gather data on pedestrian activity and tourist density, 32 students from Girne American University were positioned at central intersections along key streets in the study area (Fig 1). These intersections were carefully chosen to provide optimal vantage points for observing and measuring pedestrian flow. Each student used a camera to record short videos at their assigned locations, capturing the movement of people entering ("IN") and leaving ("OUT") the intersections. Data collection was standardized, with recordings conducted at three fixed times daily—9:00 a.m., 12:00 p.m., and 4:00 p.m.— each lasting 15 minutes to ensure consistency.

For consistency, pedestrian activity was recorded on two specific dates: May 29 (referred to as Day 29) and May 30 (Day 30). These represent a weekday and a weekend observation period, respectively. The study spanned two days to account for variations in pedestrian activity between weekdays and weekends. Data was collected on a typical working day (Monday) and a weekend day (Sunday) to compare how visitor numbers and behavior differ across these periods. Weekdays often reflect a mix of local residents and workers, while weekends typically see an increase in leisure-focused tourists. By incorporating both types of days, the research aimed to provide a comprehensive view of pedestrian dynamics within the historical city center.

Beyond the intersections, four key locations within the old city were designated as stationary observation points, chosen for their prominence as tourist attractions and urban gathering spots. Two students were stationed at each of these locations to record videos documenting the number of people present during the designated timeframes. This additional data enriched the analysis by revealing patterns of tourist congregation and activity in specific areas. The recorded videos

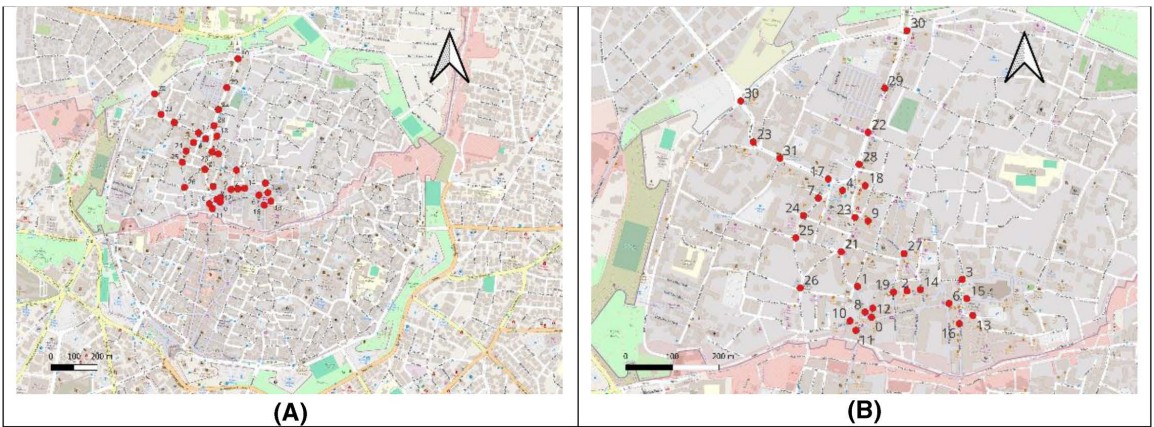

**Fig 1. The Survey locations. 32 Points are located in the old city street intersections. (A)** Survey locations across the broader urban context, highlighting the distribution of data collection points relevant to the study. **(B)** A close-up view of the specific area under investigation, providing detailed spatial context for the analyzed features and their immediate surroundings.

were analyzed to manually count the number of individuals visible at each site. The counts were then digitized using GIS MapIt, a freely available mobile application, to geo-reference the data for spatial analysis. The georeferenced data was compiled into a unified map using Carto DB, a cloud-based mapping platform, offering a detailed visualization of pedestrian activity throughout the study area. To ensure reliability, the data collection process was repeated for both observation days. This allowed for direct comparisons between weekday and weekend activity, shedding light on how temporal factors influence visitor movement and density.

The findings highlighted the spatial patterns of tourist behavior, illustrating the interplay between mobility, centrality, and the role of diverse courtyards in shaping the overall visitor experience (Fig 2). This research underscores the value of integrating digital tools, such as GIS and spatial mapping platforms, for analyzing urban dynamics. By documenting pedestrian traffic and tourist behavior across different times and days, the study provides valuable insights for urban planners, heritage managers, and tourism stakeholders. It emphasizes the importance of preserving Nicosia's cultural heritage as a cornerstone for sustainable tourism while enhancing the functionality and appeal of its historical urban spaces.

## PageRank centrality

This research employed two types of spatial centrality measures—PageRank and Straightness—to evaluate the influence of network structure on tourist flow to measure the centrality of the urban nodes in the old city of Lefkoşa, the first one is the PageRank centrality, while the second one is the closeness centrality, both are based on the spatial analysis completed by using igraph and SF network packages provided by R.

The study applied PageRank centrality analysis to evaluate the role of the street network in shaping pedestrian movement and visibility. This graph-based method is advantageous in urban analysis as it accounts for both direct and indirect connections within the network, effectively identifying influential street segments beyond their immediate neighbors. The results were integrated into GIS to visualize the distribution of central streets relative to courtyard locations.

The mathematical formula [46–49] For calculating the PageRank of a node i is

$$PR(i) = (1-d) + d \sum_{j \in M(i)}^{n} \frac{PR(j)}{L(i)} * W_{SL}$$

(1)

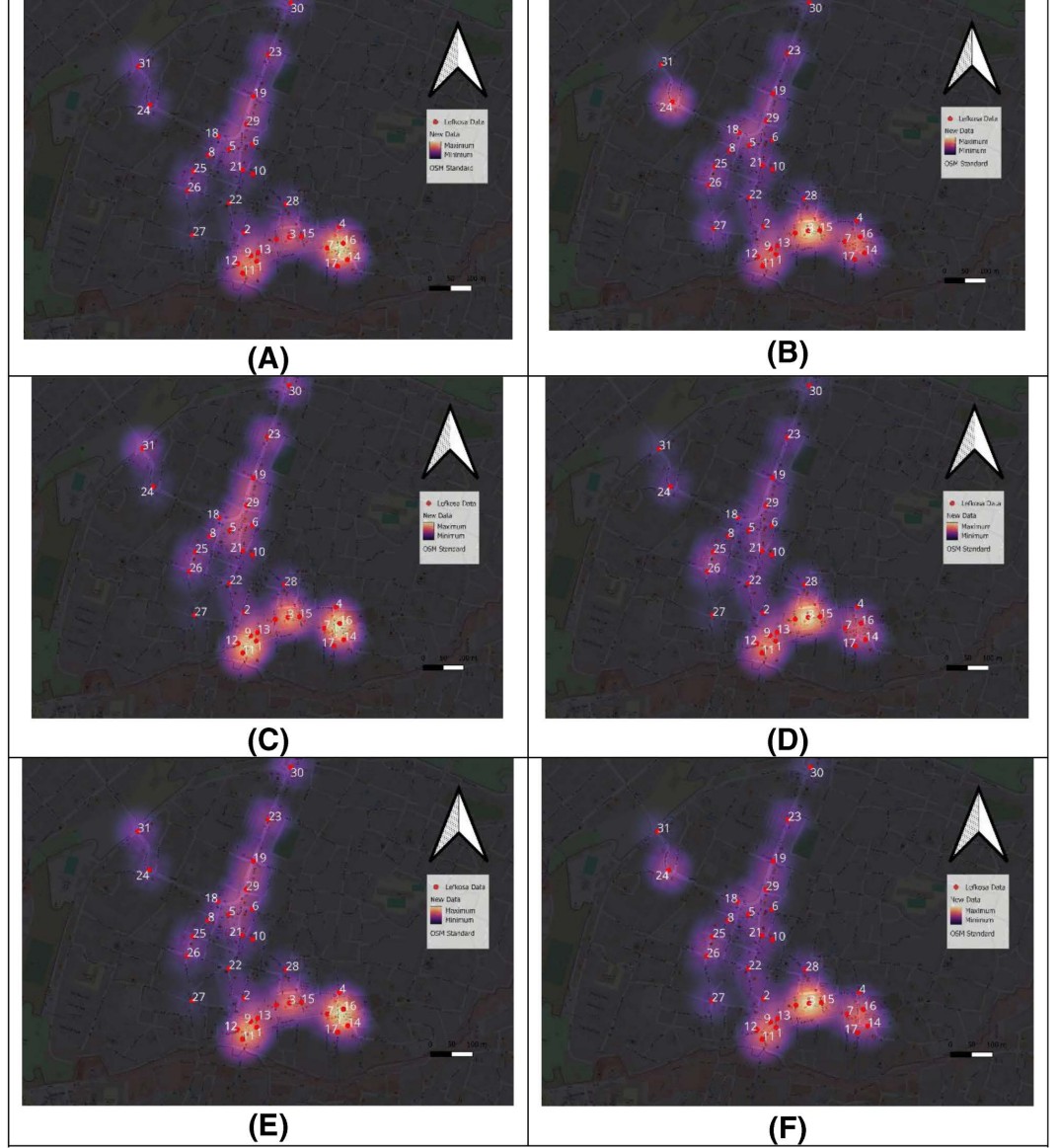

**Fig 2. The Survey Results. (A)** The number of people who entered the area on 29 May. **(B)** The number of people who entered the area was 30 May. **(C)** The number of people left the area on 29 May. **(D)** The number of people left the area on 30 May. **(E)** The Average on 29 May. **(E)** The Average on 30 May.

Where:

PR(i) denote the PageRank score of node (i).

d denotes the Damping factor (commonly set to 0.85) that accounts for the probability of continuing along a path versus restarting at a random node.

M (i) denotes the Set of nodes that link to node (i).

PR(j) denotes the PageRank score of node (j) that links to node (i).

L(j) denotes the Number of outbound links from node (j).

$W_{SL}$ denotes the weight of the street length

The integration of PageRank analysis into GIS allowed for the visualization of central streets in relation to landmark locations. This spatial representation facilitated the identification of patterns between the network's centrality and pedestrian activity.

## Straightness centrality

Straightness centrality is a critical metric in the analysis of street networks, particularly in pedestrian-oriented urban studies. It measures how efficiently a node (e.g., a street intersection) is connected to all other nodes in the network, considering the shortest possible paths. High straightness centrality indicates that a location has direct and accessible connections to other parts of the network, making it crucial for assessing walkability, accessibility, and urban connectivity. This metric helps urban planners identify key streets that facilitate efficient pedestrian movement and can inform decisions on infrastructure development or enhancements to ensure equitable accessibility. By focusing on the geometric and functional alignment of streets, straightness centrality complements other centrality measures, providing insights into the spatial coherence of the urban layout. The equation for straightness [40,50] centrality for a node v is defined as:

$$C_s(v) = \frac{1}{n-1} \sum_{u \neq v}^{n} \frac{d_E(v,u)}{d_G(v,u)}$$

(2)

$C_s(v)$ denote the Straightness centrality of node v.
n denotes the total number of nodes in the network.
$d_E(v, u)$ denote Euclidean (straight-line) distance between nodes v and u.
$d_G(v, u)$ denote the shortest path distance between nodes v and u in the network graph.
$\sum_{u \neq v}$ denote the Summation over all nodes u in the network except v.

## Generalized Linear Regression (GLR) for Poisson Distribution

Before doing the GLR, the research adopted the Moran Index to find out if the centrality of the page rank and straightness are clustered comparing tourists' density in the spatial layout. Moran Index used to figure out if the straightness, PageRank centrality and tourism density are clustered, the equation 3 used to calculate the index.

$$I = n \frac{n}{s_o} \frac{\sum_{i=1}^{n} \sum_{j=1}^{n} w_{i,j} z_i z_j}{\sum_{i=1}^{n} z_i^2}$$

(3)

Where:
$Z_i$ is the deviation of an attribute for a feature is from its mean.
$w_{i,j}$ is the spatial weight between components i and j.
The Generalized Linear Regression (GLR) [51] tool in ArcGIS Pro is widely used for modeling count data, where the dependent variable represents non-negative integer values, such as the number of events, occurrences, or incidents. The Poisson regression model is the most common form of GLR used for such cases. In Poisson regression, the dependent variable YYY is modeled as having a Poisson distribution, where the mean and variance are equal. The relationship between the predictors and the dependent variable is described by the equation

$$ln(\mu i) = \beta 0 + \beta 1 X 1 i + \beta 2 X 2 i + \ldots + \beta k X k i$$

(4)

Where:

- $\mu_i$ denotes the expected count for observation i.

- $\ln(\mu_i)$ denotes the natural log of the expected count (log link function).

- $\beta_0$ denotes the Intercept of the model (baseline log count).

- $\beta_1, \beta_2, \ldots, \beta_k$ denote the coefficients representing the effect of each independent variable.

- $X_{1i}, X_{2i}, \ldots, X_{ki}$ denote the predictor values for observation i.

The expected count for observation ($i$) is then given by:

$$\mu_i = e^{\beta_0 + \beta_1 X_{1i} + \beta_2 X_{2i} + \ldots + \beta_k X_{ki}}$$

### Integration of spatial centrality and courtyard diversity

Spatial centrality results and the landmark centrality were overlaid in GIS to explore intersections between highly central areas and landmark Regression analysis was employed to examine the relationship between centrality and tourist volume. Behavioral data from tourist observations were incorporated to contextualize the quantitative findings, revealing how centrality and diversity jointly influence visitor engagement.

In this study, courtyard "diversity" refers to the range and flexibility of functions supported within a courtyard space. Each courtyard was evaluated using a field survey checklist that captured:

(1) the number of distinct activity types observed (e.g., dining, resting, shopping, cultural events),

(2) the degree of temporal variation in usage (e.g., morning cafés vs. evening performances),

(3) user engagement patterns (e.g., passive vs. interactive), and

(4) physical adaptability (e.g., movable furniture, shaded vs. unshaded zones).

Each courtyard received a diversity score (1–5) based on a composite index summarizing these indicators. This score was used to compare courtyard performance across different spatial centrality zones. Table 1 illustrates the research methodology.

## Results

To assess spatial dependence, Moran's I was calculated on the residuals of the GLR model. The result was near zero ($I = -0.015$, $p > 0.5$), indicating no significant spatial autocorrelation and confirming the residuals were randomly distributed.

The Moran Index shows that straightness and PageRank centrality are clustered, but tourist density is scattered in the area (Fig 3).

The result of the centrality links clarified in Fig 4 with the range of the PageRank centrality (−3.570 to −2.008), while the straightness centrality range is from 0 to 0.1325, in the next step these values will be correlated with the values found in the tourist density to find the impact of these values on each other (Fig 4).

As shown in (Fig 4A), high-PageRank segments are concentrated near commercial corridors, while tourist density is more prominent in the areas shown in Fig 2B and 2F. This spatial misalignment explains the negative correlation observed in the regression model

Before model fitting, we tested for overdispersion by comparing the variance and mean of the count data. The dispersion parameter exceeded 1.5, indicating moderate overdispersion. Therefore, a negative binomial regression was also tested, but results were consistent with the Poisson GLR. Since the overdispersion was not severe, we retained the Poisson model for interpretability and consistency.

**Table 1. The Methodology of the research.**

| Process | Methodology | Approach | Calculations |
|---|---|---|---|
| Collecting Data | Tourist Behavior Data | Survey | |
| | Spatial Network Data | OSN map from ArcGIS Pro | |
| | architectural data on courtyards | Survey | |
| Spatial Data Analysis | Page Rank Analysis | Igraph, SF in R | $PR(i) = (1-d) + d \sum_{j \in M(i)}^{n} \frac{PR(j)}{L(i)} * W_{SL}$ |
| | Straightness Analysis | Igraph, SF in R | $C_s(v) = \frac{1}{n-1} \sum_{u \neq v}^{n} \frac{d_E(v,u)}{d_G(v,u)}$ |
| | Moran Index | ArcGIS Pro | $I = n \frac{n}{s_o} \frac{\sum_{i=1}^{n} \sum_{j=1}^{n} w_{i,j} z_i z_j}{\sum_{i=1}^{n} z_i^2}$ |
| | GLR | ArcGIS Pro | $ln(\mu i) = \beta 0 + \beta 1 X1i + \beta 2 X2i + \ldots + \beta k Xki$ |
| Result | Integration of Spatial Centrality and Courtyard Diversity | | |
| | Analysis and Interpretation | | |

As shown in Table 2, PageRank centrality was a significant predictor on May 29 (Day 29). Specifically, each one-unit increase in PageRank was associated with a decrease of approximately 8.7 tourist visits, showing a negative relationship with the dependent variable. Specifically, for every one-unit increase in PageRank, the dependent variable is expected to decrease by about 8.7 units. This effect is statistically significant, as evidenced by a p-value of 0.0122, which is below the conventional threshold of 0.05. The strength of this finding is reinforced by a z-statistic of −2.505. Additionally, the Variance Inflation Factor (VIF) for PAGERANK is 1.014, suggesting no concerns about multicollinearity with other variables.

The counterintuitive negative relationship between PageRank and tourist density may be due to the functional nature of central streets. In Lefkoşa, many high-PageRank streets are used for transit or administrative/commercial purposes and lack direct access to culturally active courtyards. Tourists may bypass these streets in favor of more secluded or atmospheric areas with visible and engaging courtyard activities. This suggests that spatial prominence alone does not equate to tourist attractiveness without supporting cultural or experiential value.

In contrast, **Straightness** does not demonstrate a significant relationship with the dependent variable. While its coefficient suggests a large positive effect—an increase of approximately 725.42 units for every one-unit increase in Straightness—the high standard error and corresponding z-statistic of 0.405 indicate considerable uncertainty in this estimate. The p-value of 0.685 confirms that this variable's effect is not statistically significant. Like PAGERANK, the VIF for Straightness is 1.014, showing no multicollinearity. Overall, the results suggest that PAGERANK is a meaningful factor in explaining variations in the dependent variable, while Straightness does not appear to play a significant role in this context. Both variables exhibit low collinearity, supporting the robustness of the model's findings.

On Day 30 (Table 3), the analysis reveals significant relationships between the dependent variable and two key predictors: **PAGERANK** and **Straightness**. PAGERANK exhibits a negative effect, with a coefficient of −9.02, indicating that for each one-unit increase in PAGERANK, the dependent variable is expected to decrease by approximately 9.02 units. This effect is statistically significant, as evidenced by a p-value of 0.0035, which is well below the conventional threshold of 0.05. The z-statistics of −2.922 further underscores the strength of this relationship. Moreover, the Variance Inflation Factor (VIF) for PAGERANK, at 1.014, suggests no multicollinearity concerns, reinforcing the reliability of the coefficient estimate.

Straightness also demonstrates a significant negative association with the dependent variable, with a coefficient of −4114.82. This suggests that an increase in Straightness leads to a substantial decrease in the dependent variable. The p-value of 0.0136 confirms the statistical significance of this effect, supported by a z-statistic of −2.468. Like PAGERANK,

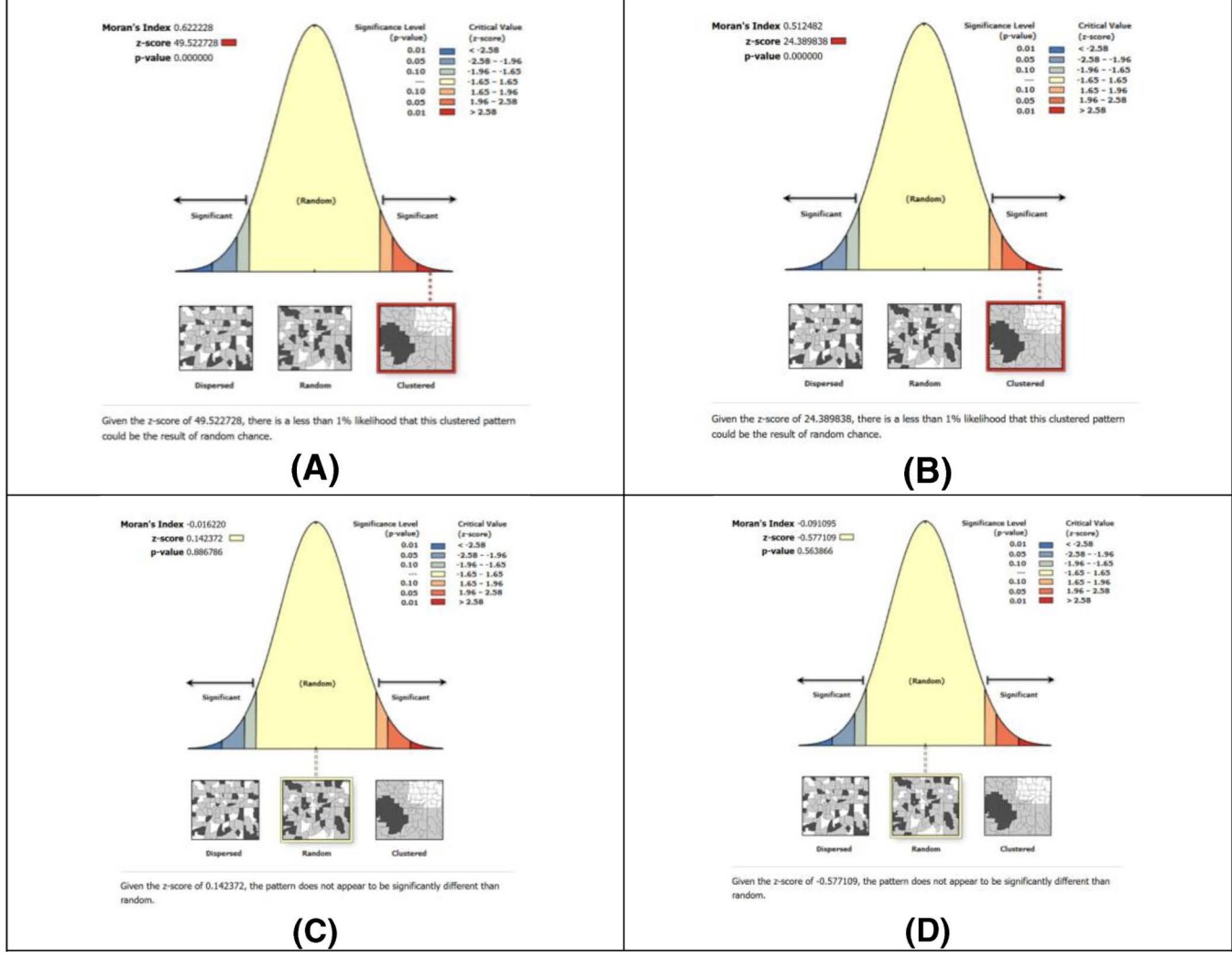

**Fig 3. Moran Index Results.** A high positive Moran's Index value of 0.922278, with a z-score of 49.52 and p-value < 0.001, indicates a highly clustered spatial pattern. **(B)** A Moran's Index of 0.512982, with a z-score of 24.39 and p-value < 0.001. **(C)** A near-zero Moran's Index of 0.016520, with a z-score of 0.14 and p-value of 0.8867. **(D)** A slightly negative Moran's Index of -0.019095, with a z-score of -0.577 and p-value of 0.5636.

the VIF for Straightness is 1.014, indicating a low risk of multicollinearity. In summary, both PAGERANK and Straightness significantly influence the dependent variable, with negative effects. These findings are statistically robust, with low multi-collinearity ensuring the stability of the model's results.

## Discussion

This study reveals that courtyards with diverse and adaptable activities play a pivotal role in enhancing tourist engagement, even in areas less centrally located. The concept of adaptability in courtyard design refers to spaces that can accommodate a variety of activities, be they cultural, recreational, social, or commercial. Such flexibility allows the courtyards to cater to different visitors' needs and preferences, thus extending their appeal over time.

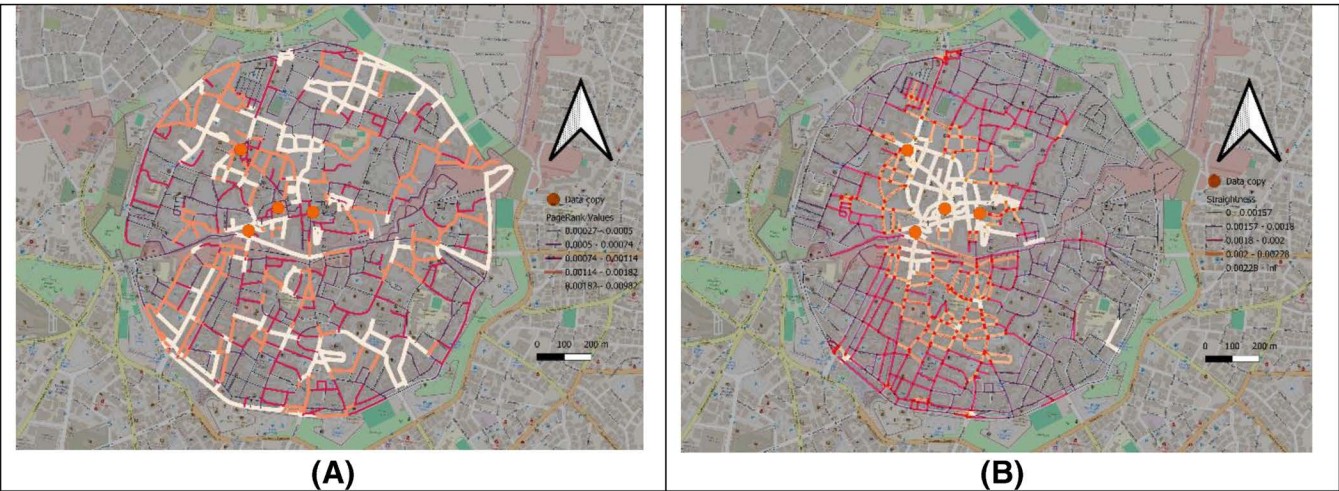

**Fig 4. Spatial Centrality Metrics. (A)** PageRank centrality values across Lefkoşa's old city: darker red lines indicate higher network influence. **(B)** Straightness centrality values: higher values highlight areas with more direct path connections. These maps help identify how spatial centrality aligns—or diverges—from areas of tourist activity.

**Table 2. Results of day 29.**

| Variable | Coefficient | Std Error | z-Statistic | p-Value | VIF |
|---|---|---|---|---|---|
| PAGERANK | −8.699225 | 3.472159 | −2.505 | 0.0122* | 1.014 |
| Straightness | 725.417013 | 1790.643060 | 0.405 | 0.685 | 1.014 |

**Table 3. Results of day 30.**

| Variable | Coefficient | Std Error | z-Statistic | p-Value | VIF |
|---|---|---|---|---|---|
| PAGERANK | −9.015502 | 3.085112 | −2.922 | 0.0035* | 1.014 |
| Straightness | −4114.818292 | 1667.445392 | −2.468 | 0.0136* | 1.014 |

Key elements contributing to the success of adaptable courtyards include:

1. The observed negative relationship between centrality metrics and tourist density may reflect hidden contextual factors in the heritage city. Many highly central streets in Lefkoşa serve as transition corridors with limited visual or functional access to courtyards, which are often concealed behind façades. Conversely, less central streets may host courtyards that are more accessible, visibly promoted, or culturally vibrant, thereby attracting more sustained visitor engagement. This suggests that spatial prominence alone does not guarantee tourist attraction unless paired with accessible and active uses

2. Multifunctionality: Courtyards designed to host a wide array of activities, such as events, markets, performances, and casual gatherings, make them more attractive to tourists. This adaptability allows them to serve diverse user needs, from solo visitors seeking a peaceful retreat to groups attending organized events.

3. Engagement through Experience: A diverse range of activities enables more immersive and longer-lasting experiences for visitors, allowing for multiple touchpoints of interaction throughout the day. This engagement increases the likelihood of return visits or extended stays in the area.

4.   Connection to Local Culture: By incorporating activities that showcase local culture, history, or traditions, courtyards become dynamic spaces where visitors can learn and experience the essence of a place in a meaningful way. This connection fosters a deeper emotional attachment to the location.

5.   Social Interaction: Multifunctional courtyards also serve as spaces for socializing, increasing the sense of community and fostering connections between tourists and locals. This human element contributes to positive visitor experiences and word-of-mouth recommendations.

6.   When comparing these results to existing literature, several key themes emerge that align with the findings of the study, while also highlighting additional insights:

7.   Spatial Centrality vs. Activity Flexibility: Previous studies have often emphasized the importance of spatial centrality in attracting tourists. Central locations tend to have higher foot traffic and visibility, which are seen as key factors in their popularity. However, this study adds to existing knowledge by showing that adaptability and diversity in courtyard activities can effectively counterbalance the disadvantages of a location's centrality. This study extends that argument by demonstrating that flexible courtyards can even thrive outside central locations when designed with diverse activities in mind.

8.   Visitor Engagement and Space Utilization: The observation that varied courtyards promote extended visitor interactions aligns with existing research on user-focused design in public spaces. Studies have highlighted that environments crafted to encourage longer stays tend to enhance the overall tourism experience. This research corroborates these findings by demonstrating that adaptable courtyard designs create opportunities for engagement, increasing the chances of visitors spending more time and making repeat visits.

9.   Cultural and Contextual Significance: Literature emphasizes the role of local culture and context in drawing tourists to space. This study builds on that perspective by demonstrating that flexible courtyards incorporating cultural elements can greatly enhance a site's attractiveness.

10.  These findings extend beyond Lefkoşa and are relevant to other walled cities and historic urban districts—such as Fez, Dubrovnik, or Mdina—where the juxtaposition of spatial centrality and hidden cultural nodes like courtyards shapes the visitor experience [52,53]. In such cities, integrating courtyard activation strategies with spatial accessibility planning can enhance tourist dispersal and engagement while avoiding over-concentration at iconic sites. Similar studies using network analysis in Córdoba (Spain) and Cairo (Egypt) have shown that spatial interventions in secondary streets and semi-private spaces improve the experiential depth of heritage tourism [53,54]. Urban planners and heritage managers can apply this model to revitalize underused areas, promoting both conservation and sustainable tourist flow

In conclusion, while much of the existing literature acknowledges the importance of spatial centrality, this study presents compelling evidence that flexible and diverse courtyards can be an equally, if not more, effective way to engage tourists. The integration of adaptive, multifunctional spaces within urban planning strategies, regardless of centrality, is crucial for fostering lasting tourist engagement.

## Conclusion

This study underscores the pivotal role adaptable and multifunctional courtyards play in enhancing tourist engagement and fostering vibrant urban spaces. For urban planners, the focus lies in designing courtyards that blend flexibility with cultural significance, transforming them into dynamic hubs for both locals and visitors. Similarly, heritage conservationists must navigate the delicate balance between preserving the historical essence of these spaces and adapting them to meet contemporary needs, ensuring their relevance while safeguarding their cultural value.

The findings highlight the importance of adaptability in courtyard design. Spaces that offer diverse and multifunctional uses, such as hosting cultural events, markets, or social gatherings, can significantly boost tourist engagement, even in less central locations. These courtyards not only attract visitors but also encourage prolonged interactions, fostering deeper connections and more meaningful experiences. While central courtyards naturally benefit from higher foot traffic, those in peripheral areas can compete effectively by offering a sense of discovery and unique, immersive experiences, revitalizing underutilized urban spaces.

Cultural integration and social interaction further enhance the appeal of courtyards. By reflecting local traditions and facilitating meaningful connections between tourists and residents, these spaces create an authentic sense of community that enriches the visitor experience. For urban planners, this means prioritizing the design of flexible, multifunctional spaces that not only alleviate congestion in central areas but also promote tourism in peripheral locations, contributing to a more balanced urban landscape.

For heritage conservationists, the challenge lies in balancing preservation with adaptability. Maintaining the architectural integrity of historic courtyards while incorporating modern, flexible functions ensures these spaces remain engaging for future generations. Additionally, designing courtyards with sustainable tourism practices in mind—such as promoting visits to less-trafficked areas—helps reduce pressure on iconic heritage sites and distributes the benefits of tourism more equitably across the city. Collaboration with local communities in the design and adaptation of courtyards fosters cultural sensitivity and inclusivity, strengthening a shared sense of ownership and pride.

Ultimately, the study demonstrates that well-designed courtyards, whether central or peripheral, can serve as catalysts for urban regeneration, cultural preservation, and sustainable tourism. By embracing multifunctionality and cultural integration, these spaces can become lasting symbols of vibrancy and connection in the urban landscape.

## Recommendations

Urban planners could create vibrant and multifunctional courtyards by integrating activity diversity with spatial centrality, making these spaces appealing regardless of their location within the city. Central courtyards naturally benefit from high visibility and heavy foot traffic, but courtyards in peripheral zones can be equally successful by offering unique and adaptable activities that encourage prolonged visitor engagement. This balanced approach ensures courtyards across the city contribute meaningfully to urban tourism while supporting sustainability and inclusivity.

Designing for flexibility is key to ensuring that courtyards can accommodate diverse uses, from cultural events and community gatherings to markets and performances. Central courtyards should leverage their visibility by hosting high-profile events, while peripheral courtyards can distinguish themselves by offering more intimate and immersive experiences, such as local craft workshops or neighborhood-specific cultural activities. By incorporating local culture and heritage into the design, these spaces can attract tourists seeking authentic and meaningful experiences, creating a deeper connection to the city's identity.

Connectivity is essential to the success of peripheral courtyards. Thoughtful planning should ensure these spaces are well-linked to public transportation, pedestrian routes, and cycling networks, encouraging visitors to explore beyond the city center. Temporary or seasonal programming, such as pop-up markets or live performances, can further enhance the appeal of peripheral courtyards, drawing tourists with unique, time-sensitive events that highlight their distinctive character.

Supporting local businesses and crafts within courtyards provides additional value, offering tourists authentic, locally produced goods while boosting the local economy. Courtyards can serve as platforms for artisans and vendors, transforming these spaces into cultural and economic hubs that benefit both visitors and residents. Promoting sustainable tourism by encouraging exploration of less central courtyards also helps distribute visitor flow more evenly across the city, reducing congestion in popular areas and fostering inclusive urban development.

Regular monitoring and evaluation of visitor experiences are crucial to refining courtyard designs and strategies over time. By analyzing tourist flow and gathering feedback, planners can continuously adapt courtyards to meet the needs of both locals and visitors. This iterative approach ensures that courtyards remain dynamic, engaging spaces that contribute to a balanced and sustainable urban tourism strategy, enriching the city's overall appeal.

## Supporting information

**S1 Data. Minimal Data.**
(ZIP)

## Author contributions

**Conceptualization:** Mustafa Aziz Amen, Farhan Abdullah Ali.

**Data curation:** Mustafa Aziz Amen.

**Methodology:** Mustafa Aziz Amen, Farhan Abdullah Ali.

**Software:** Mustafa Aziz Amen.

**Writing – original draft:** Mustafa Aziz Amen.

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
