## [Decision Letter · Decision Letter 0]

30 May 2025

PONE-D-25-07162

Analyzing the Impact of Spatial Centrality and Courtyard Diversity on Tourist Attraction in the Walled City of Lefkoşa

PLOS ONE

Dear Dr. Amen,

Thank you for submitting your manuscript to PLOS ONE. After careful consideration, we feel that it has merit but does not fully meet PLOS ONE’s publication criteria as it currently stands. Therefore, we invite you to submit a revised version of the manuscript that addresses the points raised during the review process.

Please be aware that any revised manuscript submitted with poor English, grammar, or syntax will be rejected. We strongly advise you to thoroughly review your writing or seek professional editing support if needed.Additionally, ensure that your manuscript is free from any form of plagiarism, whether intentional or unintentional, including self-plagiarism or duplication of work published elsewhere.

We look forward to receiving your revised manuscript.

Kind regards,

Tiziana Susca

Academic Editor

PLOS ONE

Journal Requirements:

3. Please include a complete copy of PLOS’ questionnaire on inclusivity in global research in your revised manuscript. Our policy for research in this area aims to improve transparency in the reporting of research performed outside of researchers’ own country or community. The policy applies to researchers who have travelled to a different country to conduct research, research with Indigenous populations or their lands, and research on cultural artefacts. The questionnaire can also be requested at the journal’s discretion for any other submissions, even if these conditions are not met.  Please find more information on the policy and a link to download a blank copy of the questionnaire here: https://journals.plos.org/plosone/s/best-practices-in-research-reporting. Please upload a completed version of your questionnaire as Supporting Information when you resubmit your manuscript.

4. In your Methods section, please provide additional information regarding the permits you obtained for the work. Please ensure you have included the full name of the authority that approved the field site access and, if no permits were required, a brief statement explaining why.

5. We note that your Data Availability Statement is currently as follows: [All relevant data are within the manuscript and its Supporting Information files

6. Please amend the manuscript submission data (via Edit Submission) to include author Farhan Abdulla Ali.

7. Please amend your authorship list in your manuscript file to include author Farhan Ali.

8. Please include a caption for figure 1.

9. We note that Figures 1, 2, 3, 4, and 5 in your submission contain map images which may be copyrighted. All PLOS content is published under the Creative Commons Attribution License (CC BY 4.0), which means that the manuscript, images, and Supporting Information files will be freely available online, and any third party is permitted to access, download, copy, distribute, and use these materials in any way, even commercially, with proper attribution. For these reasons, we cannot publish previously copyrighted maps or satellite images created using proprietary data, such as Google software (Google Maps, Street View, and Earth). For more information, see our copyright guidelines: http://journals.plos.org/plosone/s/licenses-and-copyright.

              1. You may seek permission from the original copyright holder of Figures 1, 2, 3, 4, and 5 to publish the content specifically under the CC BY 4.0 license. 

Reviewers' comments:

Reviewer's Responses to Questions

**Comments to the Author**

1. Is the manuscript technically sound, and do the data support the conclusions?

Reviewer #1: Yes

Reviewer #2: Partly

2. Has the statistical analysis been performed appropriately and rigorously? 

Reviewer #1: Yes

Reviewer #2: No

3. Have the authors made all data underlying the findings in their manuscript fully available?

Reviewer #1: Yes

Reviewer #2: Yes

4. Is the manuscript presented in an intelligible fashion and written in standard English?

Reviewer #1: Yes

Reviewer #2: No

5. Review Comments to the Author

Reviewer #1: This manuscript explores the relationship between spatial centrality and courtyard diversity in shaping tourism attraction within the walls of the historic city of Lefkosa. The topic combines urban network analysis and heritage tourism for timeliness and relevance. A mixed-methods approach - including PageRank, linear centrality, GIS-based mapping, and field observations - adds depth to the analysis. This paper makes a meaningful contribution to the urban design, spatial analysis and sustainable tourism literature. However, several areas require further clarification, reorganization, and technical improvements

1.While the abstract outlines a clear hypothesis, the introduction would benefit from explicitly framing the research questions and analytical framework. This would improve the coherence between the introduction, methodology, and discussion.

2.Some sections like Methodology and Results are overly descriptive or repetitive. The manuscript would benefit from tighter writing and clearer transitions between paragraphs. Some grammar issues and awkward phrasing (e.g., “The research used to type of spatial centrality…”) should be corrected throughout.

3. The GLR results indicate a negative relationship between PageRank/straightness and tourist density, which appears counterintuitive and contradicts initial assumptions. The authors should elaborate on this unexpected result and offer a stronger interpretation—possibly considering factors like courtyard accessibility or limitations in spatial perception.

4.The literature review (Section 2.2) spends significant space discussing PageRank in biomedical and digital networks. This could be shortened to better focus on urban applications, especially in heritage cities.

5.The figures referenced (e.g., Fig. 4–5) are not included or described in sufficient detail. All visual data should include clear legends, labels, and interpretations within the main text.

Best

Reviewer #2: Thank you for submitting your manuscript titled “Analyzing the Impact of Spatial Centrality and Courtyard Diversity on Tourist Attraction in the Walled City of Lefkoşa.” This study addresses a relevant topic at the intersection of urban spatial analysis and heritage tourism. However, the manuscript in its current form requires significant revisions before it can be considered for publication. Below I outline major and minor points that need to be addressed to improve the clarity and rigor of the paper.

Major Comments

1. Rationale for PageRank and Straightness centrality: The choice of using PageRank and Straightness centrality as measures of spatial centrality needs clearer justification. Why were these particular metrics selected, and how do they capture pedestrian movement or visibility better than other common centrality measures (e.g., betweenness or closeness)? The manuscript should explain how PageRank (a network popularity metric) and straightness (a measure related to path directness) are conceptually linked to tourist movement and attraction. Strengthen the background or methods section with an explanation (supported by appropriate references) of why these metrics are relevant for modeling tourist foot traffic in urban heritage contexts.

2. Definition and measurement of courtyard “diversity”: The idea of courtyard diversity is compelling, but the paper must clearly define what is meant by “diverse and adaptable activities” in courtyards and how this was quantified. It is currently unclear how diversity was measured – for example, does it refer to the number of different functions/events in a courtyard, the variety of amenities, historical uses, or something else? The authors should explicitly specify the criteria or index used to capture courtyard diversity (e.g., a diversity score or categories of courtyard usage). This clarification is essential for readers to understand how courtyard characteristics were assessed and to ensure that the term “diversity” is used in a consistent, measurable way throughout the analysis.

3. GLR (Poisson) model assumptions and spatial effects: The use of a Generalized Linear Regression (Poisson) model for tourists is appropriate, but several modeling details need better explanation. First, please address overdispersion: were the count data overdispersed, and if so, how was this handled (e.g., using a quasi-Poisson or negative binomial model)? Second, discuss any spatial autocorrelation in the data – for instance, did you compute Moran’s I on model residuals or tourist densities, and what were the results? If spatial effects were detected, the manuscript should describe how they were accounted for or discuss their implications on the validity of the model. Lastly, the finding that PageRank centrality had a negative relationship with tourist density is counter-intuitive and requires interpretation. Why might highly central (high-PageRank) streets see lower tourist counts? The authors should offer a plausible explanation or theory for this result (e.g., perhaps highly central streets are commercial/administrative with fewer tourist attractions, or tourists deliberately seek less central areas with interesting courtyards). Elaborating on these points will strengthen the results and their credibility.

4. Broader context and relevance of findings: The discussion would benefit from connecting the results to the broader urban tourism and heritage conservation literature. Currently, the findings are tied closely to Lefkoşa, but the authors should briefly indicate how these insights relate to similar historic city contexts. For example, can parallels be drawn to other walled cities or heritage districts where spatial network analysis has been used to explain tourist movement? Adding a couple of comparisons or citing studies from other heritage tourism sites would highlight the practical relevance and potential generalizability of the results. Additionally, the authors should emphasize how urban planners or heritage managers elsewhere could apply these findings – for instance, balancing spatial centrality improvements with activating courtyard spaces – to enhance tourist experiences in historic urban areas.

Minor Comments

1. Clarity and grammar: The manuscript would benefit from careful editing for clarity and grammar. There are a few long or unclear sentences in the Methods and Results sections that should be revised into clearer statements. For example, the phrase “Day 29” is mentioned without context – please clarify what “Day 29” refers to (is it the 29th day of observation, a label for a scenario, etc.?). Ensure that each sentence conveys a single clear idea, and avoid run-on sentences. A thorough proofreading (or copy-editing) is recommended to fix grammatical issues and improve the overall readability of the text.

2. Integration of figures into the text: The figures and maps presented in the manuscript need to be more tightly integrated with the narrative. Each figure should be explicitly referenced in the text, and its content should be described so that readers understand what to observe. For example, if Figure X shows the spatial distribution of tourist density or centrality values, mention those results in the text and explain how they support your findings. Enhancing the figure captions with more detail and ensuring that the text discusses the key insights from each figure will make the results easier to follow and more convincing.

3. Formatting and consistency: Please address some formatting inconsistencies to align with journal standards. For instance, maintain consistent capitalization for terms like “PageRank” (ensure the capital “P” and “R” are used every time, since at times it appears in lowercase). Check that mathematical notations or formulas (if any) are properly formatted (use italic or equation formatting as needed) and that all variables are defined. Additionally, ensure the citation style is consistent and adheres to PLOS ONE guidelines (e.g. verify that all references cited in the text appear in the reference list and vice versa). Cleaning up these minor formatting issues will improve the professionalism of the manuscript.

Overall, the topic of this manuscript is important and the study has the potential to contribute to urban tourism research. By addressing the points above – clarifying methods and concepts, strengthening the analysis discussion, and improving clarity and formatting – the authors can significantly improve the paper. I encourage the authors to revise accordingly and believe that doing so will make the manuscript much stronger for publication.

6. PLOS authors have the option to publish the peer review history of their article (what does this mean?). If published, this will include your full peer review and any attached files.

Reviewer #1: No

Reviewer #2: No

---

## [Decision Letter · Decision Letter 1]

8 Aug 2025

Analyzing the Impact of Spatial Centrality and Courtyard Diversity on Tourist Attraction in the Walled City of Lefkoşa

PONE-D-25-07162R1

Dear Dr. Amen,

We’re pleased to inform you that your manuscript has been judged scientifically suitable for publication and will be formally accepted for publication once it meets all outstanding technical requirements.

Kind regards,

Tiziana Susca

Academic Editor

PLOS ONE

Additional Editor Comments (optional):

Dear Authors,

I am glad to announce you that the manuscript " Analyzing the Impact of Spatial Centrality and Courtyard Diversity on Tourist Attraction in the Walled City of Lefkoşa" has been accepted for the publication in Plos One in the current form.

Thank you for submitting your article to Plos One.

Reviewers' comments:

Reviewer's Responses to Questions

**Comments to the Author**

1. If the authors have adequately addressed your comments raised in a previous round of review and you feel that this manuscript is now acceptable for publication, you may indicate that here to bypass the “Comments to the Author” section, enter your conflict of interest statement in the “Confidential to Editor” section, and submit your "Accept" recommendation.

Reviewer #2: All comments have been addressed

2. Is the manuscript technically sound, and do the data support the conclusions?

Reviewer #2: Partly

3. Has the statistical analysis been performed appropriately and rigorously? 

Reviewer #2: Yes

4. Have the authors made all data underlying the findings in their manuscript fully available?

Reviewer #2: Yes

5. Is the manuscript presented in an intelligible fashion and written in standard English?

Reviewer #2: Yes

6. Review Comments to the Author

Reviewer #2: Thank you for revising and resubmitting your manuscript titled “Analyzing the Impact of Spatial Centrality and Courtyard Diversity on Tourist Attraction in the Walled City of Lefkoşa.” The revised version shows clear improvement, and the earlier concerns have been addressed.

7. PLOS authors have the option to publish the peer review history of their article (what does this mean?). If published, this will include your full peer review and any attached files.

Reviewer #2: No

---

## [Editor Report · Acceptance letter]

PONE-D-25-07162R1

PLOS ONE

Dear Dr. Amen,

I'm pleased to inform you that your manuscript has been deemed suitable for publication in PLOS ONE. Congratulations! Your manuscript is now being handed over to our production team.

Kind regards,

on behalf of

Dr. Tiziana Susca

Academic Editor

PLOS ONE